# The Effect of Background Color on Skin Color Variation of Juvenile *Plectropomus leopardus*

**DOI:** 10.3390/ani12233349

**Published:** 2022-11-29

**Authors:** Feibiao Song, Liping Shi, Fucheng Yao, Yue Gu, Da Zheng, Weiwei Zhang, Yesong Liang, Kaixi Zhang, Min Yang, Lei Wang, Junlong Sun, Jian Luo

**Affiliations:** State Key Laboratory of Marine Resource Utilization in South China Sea, Hainan Aquaculture Breeding Engineering Research Center, Hainan Academician Team Innovation Center, College of Marine Sciences, Hainan University, Haikou 570228, China

**Keywords:** background color, MSH content, *Plectropomus leopardus*, pigments, skin color, tyrosinase activity

## Abstract

**Simple Summary:**

The integument of leopard coral grouper (*Plectropomus leopardus*) becomes black, brown, and red under intensive culture. Fish skin color is one of the most important commercial traits in aquaculture and is strongly affected by the background color. The tank colors used to rear *P. leopardus* are generally gray or blue; however, the effect of tank color on fish physiological status is poorly understood. No studies related to background color on skin color have been conducted in *P. leopardus*. To further understand the molecular mechanisms of skin pigmentation in *P. leopardus*, fifteen experimental aquaria with circulating water were prepared, and 12 aquaria were pasted with the labels Blue, Red, Black, or White on opaque polypropylene plastic board. Three aquaria were Transparent. The results showed that lighter colors inhibited the formation of melanocytes and had a significant effect on carotenoid and lutein contents. Pigment-related genes were involved in the regulation of fish skin color and were affected by background color in *P. leopardus*. These results indicate that a white background is more conducive to maintaining red skin color in juvenile *P. leopardus*. Our findings provide a new idea on the culture of *P. leopardus*.

**Abstract:**

Fish skin color is usually strongly affected by the background color of their environment. The study investigated the effects of five different background colors on the skin color of leopard coral groupers (*Plectropomus leopardus*). More than 450 juveniles were reared in Blue, Red, Black, White, and Transparent background tanks for 56 days. The paraffin section showed that the skin melanin zone of fish in the White group was smaller, whereas the Black and Red groups (especially Black) were nearly the largest. The apparent skin color of *P. leopardus* was red on the white background, which darkened in response to the other color backgrounds. The Black group revealed the blackest skin color, followed by the transparent group. Moreover, the White group had the highest L*, a*, and b* values. The melanin content and tyrosinase activity in the dorsal and ventral skin of the Black group were significantly higher than those in the other groups (*p* < 0.05), and the serum α-MSH level was higher in the Black group as well. The carotenoid and lutein contents showed completely different trends among the experimental groups, as carotenoid content was higher in the Red and White groups, while lutein content was higher in the Transparent group. The expression level of *scarb*1 was highest in the Blue and White groups, followed by the Transparent group, and lowest in the Black group (*p* < 0.05). The expression trend of *scarb*1 was similar to the skin color in different backgrounds, indicating that the background color regulated *scarb*1 expression level through visual center, then influenced the uptake and transport of carotenoids, then influenced the skin color formation of *P. leopardus*. Moreover, lighter colors inhibited the formation of melanocytes and had a significant effect on carotenoid and lutein contents. Pigment-related genes were involved in the regulation of fish skin color, and they were affected by background color in *P. leopardus*. These results indicate that a white background is more conducive to maintaining red skin color in juvenile *P. leopardus*.

## 1. Introduction

Fish skin color is one of the most important commercial traits in aquaculture. Skin color has been classified into morphological and physiological types, both of which are affected by the interaction between environmental factors and genetics [1,2,3,4,5]. Teleosts can change their skin color or hue quickly by translocating melanosomes within the skin chromatophores, depending on differences in the light intensity and background color [5,6]. The color change is closely associated with neuroendocrine and endocrine systems in the hypothalamus–pituitary axis [7,8].

Five different pigment cell types have been identified in teleosts, including melanocytes, xantho-/erythrophores, iridophores, white leucophores, and blue cyanophores, and the color of the fish body is determined by the changes in pigment cells [9]. In many teleosts, adaptation to the background occurs as a physiological response, including aggregation and dispersion of pigments triggered by neural stimuli [8,10]. The apparent body color of Malaysian red tilapia is paler on a white background and darker on a black background, and the body color varies in response to transfer to the opposite background color [11]. Bright light causes aggregation of melanosomes, leading to a pale skin color, and dim light disperses the melanosomes [12]. In *Oryzias latipes*, the density of chromatic nerve fibers changes along with changes in the number of melanophores during prolonged background adaptation; melanophore size decreases first, followed by a decrease in the density of melanophores caused by gradual cell death on a white background [13,14]. *Oreochromis niloticus* exhibits a high cortisol level when maintained on blue and brown backgrounds [15], and fish reared on a black background are distinctively darker than those reared on a white, blue, or clear background [16,17,18]. Furthermore, the background color could influence the capture of food and consumption, and consequently, the growth of the fish [19,20,21].

Melanin-concentrating hormone (*mch*) and α-melanophore-stimulating hormone (α-MSH) are two peptide hormones controlling body color with opposite functions in the chromatophores of fish [7,8,22]. Body color in *Oncorhynchus mykiss* is affected by tank brightness; fish in white tanks have the brightest body color and the highest *mch* expression levels, and proopiomelanocortins (*pomc*-a and *pomc*-b) are more highly expressed in black tanks [23]. The body color of *Carassius auratus* exposed to fluorescent light on a white background is paler than that of fish held on a black background [23]. The aggregation of pigment induced by adapting to a white background has been associated with increased *mch* expression [24]. Recent research has shown that the expression levels of prepro-melanin-concentrating hormone 1 (*pmch*1) and proopiomelanocortin (*pomc*) are affected by the background color in *C. auratus* [25]. Moreover, carotenoids are also one of the main pigments that affect the color characteristics of fish. In *Dawkinsia filamentosa*, the black background tank helped body pigmentation and promoted carotenoid accumulation [26]. In addition, the carotenoid content was regulated by *scarb*1 (Scavenger receptor class B type I) [27,28]. The *scarb*1 gene is involved in the selective absorption and transport of carotenoids, and this function is conservative among different species [29].

The leopard coral grouper (*Plectropomus leopardus*) is a valuable marine fish in the family Epinephelidae. Moreover, this species is an important resource for intensive industrial farming in recirculating aquaculture systems due to their high nutritional value, tender flesh, beautiful skin color, and high breeding density. *P. leopardus* has high commercial value and broad market prospects. The integument of this fish becomes black, brown, and red under intensive culture [30], which is an important factor when determining fish quality, as Chinese markets prefer fish with a bright red skin color. Research has shown that different proportions of carotenoids change the skin color in this species [30]. The tank colors used to rear *P. leopardus* are generally gray or blue; however, the effect of tank color on fish physiological status is poorly understood. No studies related to the effect of background color on body color have been conducted in *P. leopardus*. To further understand the molecular mechanisms of skin pigmentation in *P. leopardus*, this study investigated the effects of background color on skin color and the endocrine system concerning body color changes and metabolism. This is the first study to examine the skin changes associated with the background color of *P. leopardus* and provide important guide for breeding of red skin color of *P. leopardus*, which will help to establish a new method of skin color regulation.

## 2. Materials and Methods

### 2.1. Ethics Approval and Consent to Participate

All experiments were performed according to the Guidelines for the Care and Use of Laboratory Animals in China. All experimental procedures and sample collection were approved by the Institutional Animal Care and Use Committee of the College of Ocean of Hainan University, Hainan, China (protocol code HNUAUCC-2021-00007, 26 February 2021).

### 2.2. Fish

The experimental fish were obtained from the Dongfang Star Technology Co., Ltd. (Ledong, Hainan Province, China). Juvenile fish were about 10 g at 4 months post-hatch. They were maintained in experimental culture facilities during the acclimation and experimental periods under a 12 h natural photoperiods light/dark cycle at 28 ± 1 °C, pH range at 7.5~8.0, NH_4_—N < 0.5 mg/L, NO_2_—N < 0.1 mg/L, NO_3_—N < 12 mg/L. Aeration was supplied to each tank for 24 h and about a quarter of water was exchanged per day, and fish fed three times daily with a compound feed (Guangdong Yuequn Biotechnology Co., Ltd., Jieyang, China).

### 2.3. Experimental Tanks

Fifteen experimental aquaria with circulating water (length × width × depth = 60 × 40 × 40 cm) were prepared, and 12 aquaria were pasted with the labels Blue (Blue-1, Blue-2, Blue-3), Red (Red-1, Red-2, Red-3), Black (Black-1, Black-2, Black-3), or White (White-1, White-2, White-3) on opaque polypropylene plastic board. Three aquaria were Transparent (Transparent-1, Transparent-2, Transparent-3) (Appendix A). Fishing nets were placed over the aquaria to prevent the fish from escaping.

### 2.4. Experiment

These juveniles *P. leopardus* were randomly divided into five groups of Blue, Red, Black, White, or Transparent, with three replicates in each group. At the beginning of the experiment, we randomly chose 10 fish to measure the L* (lightness), a* (redness), and b* (yellowness) values of the dorsal, ventral, head, and caudal peduncle skin. Photographs were taken of the three fish. Then, healthy acclimated fish (average initial body weight 10.5 g) were randomly stocked into the 15 experimental aquaria with 30 fish per aquarium on 23 November 2020. The fish were reared in the indoor aquaria (a room with transparent roof) for 56 days from 23 November 2020 to 17 January 2021 and fed a commercially prepared diet (Guangdong Yuequn Biotechnology Co., Ltd., Jieyang, China) to satiation three times daily at about 08:00, 12:00, and 17:00.

### 2.5. Fish Sampling and Tissue Preparation

At the end of the experiment, photographs were taken of three fish per group. We randomly chose 10 fish from each group to measure the L*, a*, and b* values of the dorsal, ventral, head, and caudal peduncle skin with the ColorQuest XE (HunterLab, Reston, VA, USA). Nine fish were randomly selected from each group with three fish in each replicate. Then, blood samples were taken from the caudal vein with syringes, and serum samples were obtained for tyrosinase activity and α-MSH analyses after centrifugation (3000 g for 15 min) at 4 °C. Tissue samples, including dorsal and ventral skin of the fish from each group, were collected, snap-frozen in liquid nitrogen, and stored at −80 °C until processed.

### 2.6. Determination of Pigments, MSH Content, and Tyrosinase Activity

Nine fish per group were examined for the contents of skin melanin, carotenoids, lutein, and α-MSH, and serum α-MSH and tyrosinase activity were measured by ELISA kit (Zhenke Industrial International, Zhongshan, China). The skin was washed with precooled PBS (0.01 M, pH = 7.4), weighed, homogenized with cold PBS, and centrifuged (2000 g for 20 min) to obtain the supernatant for determination. The standard wells followed the test sample wells and the blank wells on the antibody-coated plate. Various 50 μL aliquots of different concentrations of the standard solution were added to each well. A 40 μL aliquot of the Sample Diluent and 10 μL of sample were added to each sample well. A 100 μL aliquot of HRP Conjugate was added to all wells except the blank well. The plate was covered and incubated at 37 °C for 60 min. The liquid in the wells was discarded and washing buffer was added to each well and allowed to stand for 30 s. Then, the liquid was discarded. These steps were repeated five times. A 50 μL aliquot of Chromogen Solution A and 50 μL of Chromogen Solution B were added to each well, mixed, and held at 37 °C for 15 min in the dark. A 50 μL aliquot of Stop solution was added to each well to stop the chromogenic reaction. The solution in the wells changed from blue to yellow. The absorbance of each well was read at 450 nm within 15 min after stopping the reaction. The zero was set with the blank well.

### 2.7. RNA Extraction and Quantitative Real-Time Polymerase Chain Reaction

Total RNA was extracted from dorsal and ventral skin using TRIZOL (Invitrogen, Carlsbad, CA, USA) according to the manufacturer’s protocol, and nine fish per group were used. Genomic DNA was removed from the RNA samples using DNase I (New England Biolabs, Ipswich, MA, USA). The concentration of total RNA was measured with a UV spectrophotometer (NanoDrop 2000, Thermo Scientific, Waltham, MA, USA) and quality and integrity were checked at an OD of 260/280 by 1% agarose gel electrophoresis. First-strand cDNA was synthesized using the PrimeScript RT Master Mix (Takara, Shiga, Japan), and qPCR was performed on the ABI PRISM 7500 Real-time PCR System (ABI, Foster City, CA, USA). The amplification reactions were performed in a total volume of 25 μL, including 12.5 μL of 2× SYBR Green MasterMix reagent, 1 μL of cDNA, 1 μL of each primer (10 μM), and 9.5 μL of PCR-grade water. The thermal cycling profile consisted of initial denaturation at 95 °C for 5 min followed by 40 cycles of denaturation at 95 °C for 15 s and annealing/extension for 45 s at 60 °C. The details are shown in Song et al. [31]. All primers were designed using Primer Premier 5 (Table 1).

### 2.8. Statistical Analysis

All data are presented as mean ± standard error and were analyzed using SPSS version 22.0 software (SPSS Inc., Chicago, IL, USA). All data was analyzed by one-way ANOVA after a homogeneity of variance test. A *p*-value < 0.05 was considered significant. Duncan’s multiple range tests were used to identify differences between experimental groups when significant differences were found. Comparisons between the two groups were performed using Student’s *t*-test.

## 3. Results

### 3.1. Effects of Background Color on Skin Color

The skin color of *P. leopardus* reared in this study is shown in Figure 1. At the end of the experiment, the White group maintained a red skin color compared to the Initial group, followed by the Blue group. Nevertheless, the skin color of the fish in the Red, Black, and Transparent groups blackened, followed by that in the Transparent group.

### 3.2. Slice Microstructure Observations of the Skin

Paraffin sections of the dorsal and ventral skin were prepared to observe the effect of background color on skin pigment cells (Figure 2). In the Initial group (Figure 2A), the melanin zone in the dorsal and ventral skin was smaller than that in the other groups. The melanin zone of the fish skin in the White group was smaller, whereas that in the Black and Red groups was nearly the largest (Figure 2C,D). The melanin zone on the dorsal skin was larger than that of the ventral skin in all groups. Interestingly, the melanin granules on the dorsal skin were almost at the surface of the skin in the Transparent group (Figure 2F3).

### 3.3. Effects of Background Color on the L*, a*, and b* Values

The L*, a*, and b* values of the dorsal, ventral, head, and caudal peduncle skin of *P. leopardus* are shown in Table 2 and Table 3. As shown in Table 2, the average L* value of the ventral skin was higher than that of the dorsal skin (*p* < 0.05). The L* value of fish skin in the White group was the highest among all experimental groups (*p* < 0.05), whereas that in the Black group was the lowest (*p* < 0.05). Moreover, the trend in the a* value of the dorsal skin and the b* values of the dorsal and ventral skin were similar to the L* value. Nevertheless, the a* value of the ventral skin was highest in the White group among all experimental groups (*p* < 0.05), whereas the a* value of the Red group was the lowest (*p* < 0.05). In Table 3, no significant differences in the L* or a* values were observed between the head skin and caudal peduncle skin. The b* values of the head skin were higher than those of the caudal peduncle skin in the Initial and White groups (*p* < 0.05); however, no significant differences were detected among the other experimental groups.

### 3.4. Pigment Content and Tyrosinase Activity in Fish Skin

The pigment content and tyrosinase activity in fish skin from the experimental groups are shown in Figure 3. The melanin contents in the dorsal and ventral skin of the Black group were significantly higher than those in the other groups (*p* < 0.05), whereas the White and Blue group levels were the lowest (*p* < 0.05) (Figure 3A). In addition, the trends in tyrosinase activity were similar to the trends in melanin content among the groups (Figure 3B). Interestingly, carotenoid and lutein contents had completely different trends among the experimental groups. The carotenoid content was highest in the dorsal skin from the Red and White groups (*p* < 0.05) (Figure 3C). The ventral skin from the White group had the highest carotenoid content (*p* < 0.05), and the Black group had the lowest (*p* < 0.05) (Figure 3C). Lutein content was lower in the White and Black groups than the other groups (*p* < 0.05), and the Transparent group had the highest lutein content (*p* < 0.05) (Figure 3D).

### 3.5. Serum Tyrosinase Activity and α-MSH Level

As shown in Figure 4, the Black group had the highest serum tyrosinase activity (*p* < 0.05), followed by the Red and Transparent groups. The serum tyrosinase activity of the Blue and White groups was the lowest (*p* < 0.05) (Figure 4A). The serum α-MSH levels were similar to those of tyrosinase activity except in the Transparent group (Figure 4B).

### 3.6. Expression of Fish Skin Color-Related Genes

The results of fish skin melanin and carotenoid biosynthesis-related gene expression in the skin of *P. leopardus* are shown in Figure 5. The expression levels of *mch* and *pomc* in *P. leopardus* skin were significantly higher in the Red group (*p* < 0.05), followed by the Transparent group (*p* < 0.05). No significant differences in the *mch* or *pomc* expression levels were observed in the Blue, Black, or White groups (Figure 5A,B). The *tyr* expression level was highest in the Black group (*p* < 0.05), followed by the Red and Transparent groups (Figure 5C). The *scarb*1 expression level was highest in the Blue and White groups, followed by the Transparent group. *scarb*1 expression was lowest in the Black group (*p* < 0.05) (Figure 5D).

## 4. Discussion

The skin color of *P. leopardus* is typically red; however, skin of most fish will turn black under artificial culture conditions, which affects their economic value [32]. Fish skin color is affected by the nervous and endocrine systems, as well as changes in nutrition and the environment. Here, we investigated whether the skin color of *P. leopardus* was affected by the background color of the tanks in which the fish were held. Studies have shown that background color can affect the skin color variation of aquatic animals [33].

Fish skin pigmentation or body color is one of the most important quality criteria affecting the economic value of fish for human consumption and ornamental use [34,35]. In this study, the apparent skin color of *P. Leopardus* was red on a white background, which darkened when the fish were held on the other color backgrounds. The Black group had the blackest skin color, followed by the Transparent group (Figure 1), which was consistent with the findings for *C. auratus* [12,24], red tilapia [11], and other fish [33]. Paraffin sections of the dorsal and ventral skin showed that darkening of the fish skin was caused by an increase in the number of melanocytes in the experimental groups (Figure 2). Paler colors indicated fewer melanocytes, and this was likely caused by the visual opsin perception of the tank colors [36].

The L*, a*, and b* values have been frequently used to quantify fish color [33,37,38]. The White group demonstrated the highest L*, a*, and b* values in the dorsal, ventral, head, and caudal peduncle skin of *P. leopardus* in this study (Table 2 and Table 3), which was consistent with the brighter, paler, more yellow and red skin color on the White background than that on the other color backgrounds. During our study, the fish subjected to the treatments with dark backgrounds had a deeper melanin pigmentation, which could have resulted from their camouflage mechanism and an attempt to simulate the environmental colors [39].

Fish body coloration is the result of diverse pigments synthesized by pigment cells or chromatophores [9]. Here, carotenoid and lutein contents showed completely different trends than melanin. Carotenoid content was higher in the Red and White groups, while lutein content was higher in the Transparent group (Figure 3). This is inconsistent with the results of *C. auratus* and *D. filamentosa*, which may be caused by different species [18,26]. Previous studies have shown that the coloring effect of carotenoids is affected by the distribution of melanocytes [29]. This result suggests that carotenoids and other pigments, such as lutein and melanin, regulate fish skin color. However, fish cannot synthesize carotenoids by themselves, as these come from algae and other foods [29]. Although carotenoid-based coloration is dietary-dependent, genetic factors also play an indispensable role in carotenoid pigmentation. *scarb*1 is a key gene directly related to carotenoid coloring in vertebrates that mediates carotenoid absorption and transportation [40]. Here, the *scarb*1 expression level was highest in the Blue and White groups, followed by the Transparent group, and lowest in the Black group (*p* < 0.05), showing the same trend as skin color (Figure 5D). Specific wavelengths could regulate the fish skin color through neuropeptide hormones and photoreceptors [41]. Our results suggest that the background color regulated *scarb*1 expression level through the visual center, then influenced the uptake and transport of carotenoids, then influenced the skin color formation of *P. leopardus*. These results indicate that the White background was more conducive for juvenile *P. leopardus* to maintain their red skin color.

Tyrosinase is a rate-limiting enzyme in the melanin synthetic pathway, and its activity plays an important role in regulating fish skin color. A correlation has been reported between tyrosinase activity and melanin content in fish, and it is affected by genetic, nutritional, and environmental conditions, as well as by the developmental stage of the fish [42,43,44,45,46]. The trend in tyrosinase activity and melanin content in skin of *Cyprinus carpio* is consistent with different skin colors [46]. In this study, the melanin content and tyrosinase activity in the dorsal and ventral skin of the Black group were significantly higher than those in the other groups (*p* < 0.05), and the highest *tyr* expression level was detected (Figure 3A,B). Therefore, it can be considered that when fish were reared in the dark background, the tyrosinase activity was increased, which stimulated the production of melanin pigment in the skin. In addition, the melanin synthesis affected the variations in fish skin color, and its content reflects differences in skin color.

In teleosts, chronic treatment with α-MSH darkens body color [6,47]. Injecting red tilapia with α-MSH in the caudal vein results in significantly higher tyrosinase activity and melanin content in the dorsal and ventral skin [48]. In this study, the Black group had the highest serum α-MSH level, followed by the Red group. A similar trend was observed for serum tyrosinase activity (Figure 4). This is consistent with previous results, indicating that the α-MSH level can be affected by background color and that a dark color increases the α-MSH level. α-MSH is a peptide derived from *pomc* and *mch* that is competitively involved in body color regulation. α-MSH induces a dark body color, while Mch induces a pale body color [25]. Interestingly, the expression levels of *pomc* and *mch* in skin were highest in the Red group, followed by the Transparent group, while the Black group expressed the lowest levels of *pomc* and *mch* in this study (Figure 5). These results suggest that the red background had a stronger effect on *pomc* and *mch* gene expression than the other colors.

## 5. Conclusions

In this study, the apparent skin color of *P. leopardus* was red in a white background, which darkened in response to the other color backgrounds. The Black group exhibited the darkest skin color, followed by the Transparent group. The darkening of fish skin was caused by an increase in the number of melanocytes in the experimental groups. Moreover, the White group had the highest L*, a*, and b* values, which was consistent with the brighter, paler, more yellow and red skin color on the White background than that on the other color backgrounds. The melanin content, tyrosinase activity, and serum α-MSH level in the dorsal and ventral skin of the Black group were significantly higher than those in the other groups (*p* < 0.05). The carotenoid content was higher in the Red and White groups, while lutein content was higher in Transparent group. Moreover, the paler colors inhibited the formation of melanocytes and had a significant effect on carotenoid and lutein contents. Pigment-related genes are involved in the regulation of fish skin color, and they are affected by background color in *P. leopardus*. Therefore, a white background is more conducive for juvenile *P. leopardus* to maintain red skin color.

## Figures and Tables

**Figure 1 animals-12-03349-f001:**
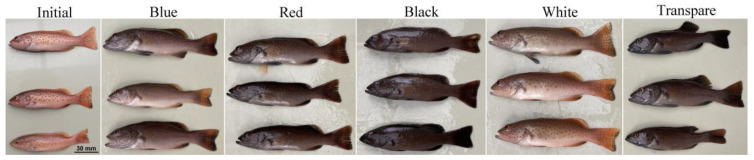
The apparent skin color of *P. leopardus* rearing in different background tanks.

**Figure 2 animals-12-03349-f002:**
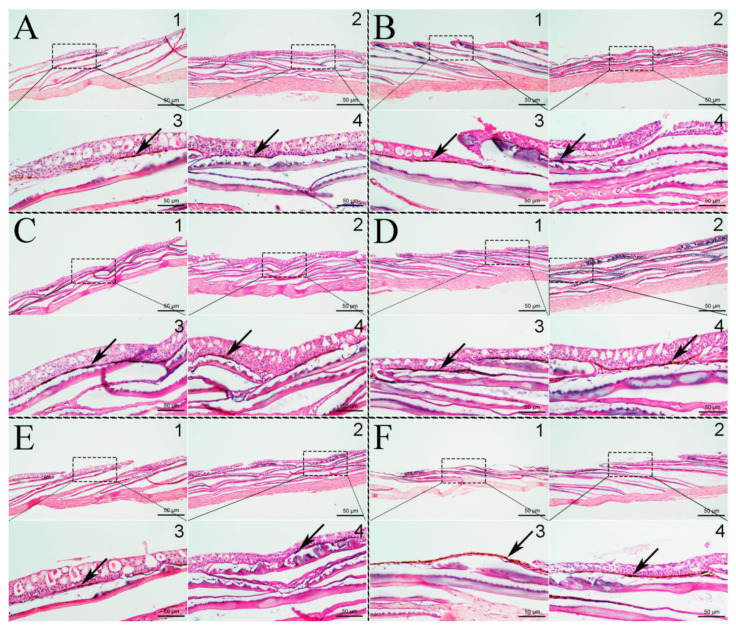
Slice microstructure observation of melanin zone in dorsal and ventral skin of *P. leopardus*. (**A**): Initial group; (**B**): Blue group; (**C**): Red group; (**D**): Black group; (**E**): White group; (**F**): Transparent group. Black arrows indicate melanin. In the subfigures, 1 is the dorsal skin and 2 is the ventral skin. 3 is the local amplification of 1, and 4 is the local amplification of 2.

**Figure 3 animals-12-03349-f003:**
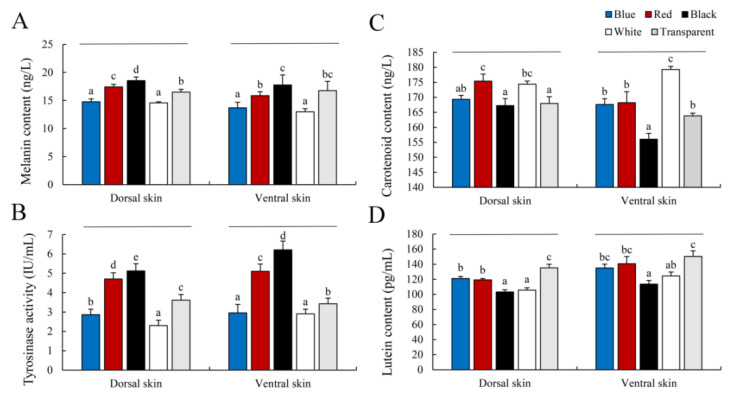
Pigment content and tyrosinase activity in skin. Values are means ± standard error for ten replicates. (**A**): Melanin content; (**B**): Tyrosinase activity; (**C**): Carotenoid content; (**D**): Lutein content. Significant differences for means within experimental groups are indicated with different lowercase letters (*p* < 0.05, n = 9).

**Figure 4 animals-12-03349-f004:**
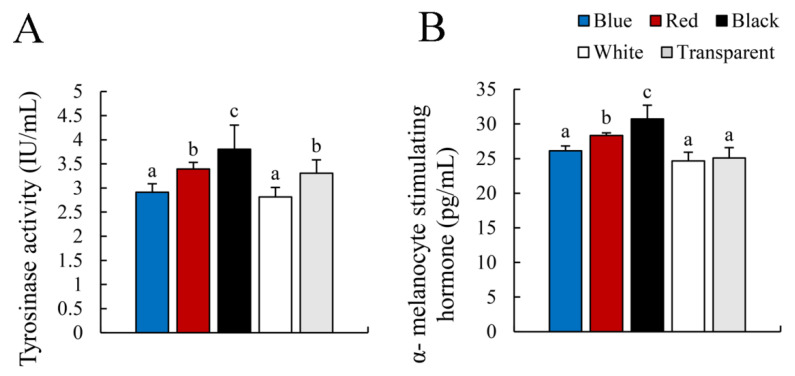
Tyrosinase activity and α-MSH levels in serum. (**A**): Tyrosinase activity; (**B**): α-melanocyte stimulating hormone. Values are means ± standard error for ten replicates. Significant differences for means within experimental groups are indicated with different lowercase letters (*p* < 0.05, n = 9).

**Figure 5 animals-12-03349-f005:**
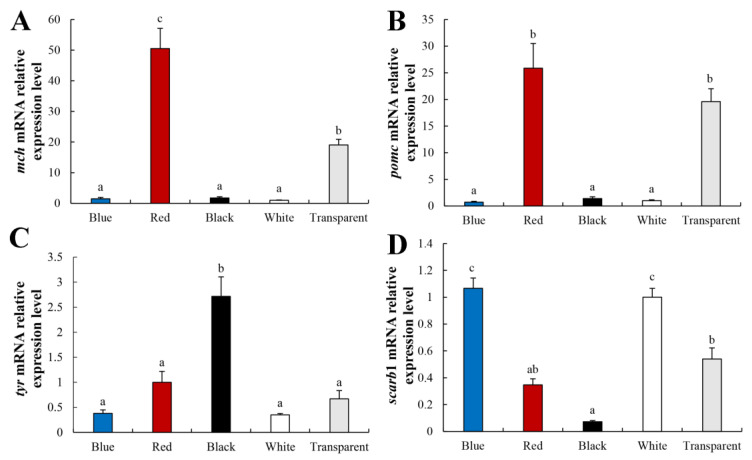
The expression levels of *mch* (**A**), *pomc* (**B**), *tyr* (**C**), and *scarb*1 (**D**) in *P. leopardus* skin for different color. Values are means ± standard error for ten replicates. Significant differences for means within experimental groups are indicated with different lowercase letters (*p* < 0.05, n = 9).

**Table 1 animals-12-03349-t001:** Primer sequences.

Primer		Sequences (5′–3′)
*tyr*	F	GGTCGCATAGACAGTGCTTCC
	R	GTCTTCAACATCCTCAGCGGT
*mch*	F	TGCTCTGTCAGTGGCGATAC
	R	GAGGGACAGTCCGTTGTGTT
*pomc*	F	AGTCAGTGCTGGGAACATCC
	R	GTCGAGATCTGACGGAGGAG
*scarb*1	F	CACCGTGTCCTACAGGGAGT
	R	ACCAGTCCGCTGTCATAACC
*β-actin*	F	CACCACAGCCGAGAGGGA
	R	TCTGGGCAACGGAACCTCT

**Table 2 animals-12-03349-t002:** Effects of background color on the L*, a*, and b* values of the dorsal and ventral skin.

	Dorsal Skin			Ventral Skin		
	L*	a*	b*	L*	a*	b*
Initial	37.95 ± 1.22 ^Ae^	5.34 ± 0.20 ^d^	6.20 ± 0.37 ^Ad^	52.73 ± 0.50 ^Bd^	6.10 ± 0.37 ^c^	9.40 ± 0.46 ^Bd^
Blue	29.10 ± 0.63 ^Ac^	1.92 ± 0.14 ^Ab^	3.57 ± 0.18 ^c^	42.93 ± 2.26 ^Bc^	3.85 ± 0.50 ^Bb^	3.78 ± 0.48 ^c^
Red	25.90 ± 0.84 ^Ab^	0.80 ± 0.09 ^a^	2.27 ± 0.20 ^Ab^	37.00 ± 1.66 ^Bb^	1.13 ± 0.15 ^a^	3.32 ± 0.30 ^Bb^
Black	22.62 ± 0.46 ^Aa^	0.70 ± 0.06 ^Aa^	1.32 ± 0.16 ^Aa^	31.62 ± 1.33 ^Ba^	1.63 ± 0.17 ^Ba^	2.67 ± 0.28 ^Ba^
White	33.45 ± 0.88 ^Ad^	2.57 ± 0.25 ^c^	4.23 ± 0.24 ^c^	51.33 ± 2.23 ^Bd^	3.72 ± 0.27 ^b^	4.45 ± 0.11 ^d^
Transparent	26.00 ± 0.85 ^Ab^	0.85 ± 0.08 ^Aa^	2.32 ± 0.18 ^b^	31.60 ± 1.24 ^Ba^	1.52 ± 0.16 ^Ba^	2.50 ± 0.23 ^a^

Note: Values are means ± standard error for ten replicates. Significant differences for means within experimental groups are indicated with different lowercase letter superscripts (*p* < 0.05, n = 10); Significant differences between means in the dorsal skin and ventral skin groups are indicated with different capital letter superscripts (*p* < 0.05, n = 10).

**Table 3 animals-12-03349-t003:** Effects of background on the L*, a*, and b* values of the head and caudal peduncle skin.

	Head Skin			Caudal Peduncle Skin		
	L*	a*	b*	L*	a*	b*
Initial	46.45 ± 0.44 ^d^	7.44 ± 0.29 ^c^	8.75 ± 0.26 ^Ac^	43.49 ± 0.59 ^d^	6.39 ± 0.18 ^c^	6.37 ± 0.29 ^Bc^
Blue	36.35 ± 1.13 ^b^	3.55 ± 0.42 ^b^	4.70 ± 0.38 ^b^	36.13 ± 0.56 ^c^	3.53 ± 0.37 ^b^	4.08 ± 0.21 ^b^
Red	31.57 ± 0.87 ^a^	1.27 ± 0.15 ^a^	2.40 ± 0.28 ^a^	32.02 ± 0.93 ^b^	1.55 ± 0.20 ^a^	2.73 ± 0.22 ^a^
Black	30.42 ± 0.83 ^a^	1.20 ± 0.13 ^a^	1.78 ± 0.07 ^a^	28.03 ± 0.99 ^a^	1.68 ± 0.19 ^a^	2.25 ± 0.18 ^a^
White	40.35 ± 1.14 ^c^	3.92 ± 0.42 ^b^	5.13 ± 0.50 ^Ab^	42.48 ± 1.81 ^d^	3.70 ± 0.29 ^b^	3.73 ± 0.50 ^Bb^
Transparent	30.88 ± 1.20 ^a^	1.23 ± 0.42 ^a^	1.94 ± 0.16 ^a^	27.58 ± 0.72 ^a^	1.62 ± 0.10 ^a^	2.30 ± 0.19 ^a^

Note: Values are means ± standard error for ten replicates. Significant differences for means within experimental groups are indicated with different lowercase letter superscripts (*p* < 0.05, n = 10); Significant differences for means between head skin and caudal peduncle skin groups are indicated with different capital letter superscripts (*p* < 0.05, n = 10).

## Data Availability

Not applicable.

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
