# Peer review of "The Effect of Background Color on Skin Color Variation of Juvenile Plectropomus leopardus"

_animals, 2022, doi:10.3390/ani12233349_

Round 1

Reviewer 1 Report

The authors investigated the effect of background color on skin colour variation of juvenile Leopard coral grouper Plectropomus leopardus. They designed five treatments to test their hypothesis. This manuscript (MS) was clearly written and easy to understand. This work can help the sustainability of this species farming as few studies have been done on this topic. However, some minor issues significantly compromised the quality of this MS.

However, I have touched on some more points that can contribute to the improvement of this MS.

Minor comments

·       Line 72, and throughout the MS, please first mention the common name plus scientific name, and for the rest of the MS, just report the common name.

·       You wrote the next sections very well

·       Here and elsewhere, report P uppercase and italic (P<0.05).

·       Throughout the MS, if there is no significant difference, no need to report P-value.

·       Please reorder the keywords alphabetically and capitalize each word.

·       Well-developed introduction and included a clear fellow and relevant points.

·       Please update the introduction with recent works as many studies are available from the last two years, which were not included in this section.

·       Please mention the novelty of your work in the last paragraph of the introduction.

·       Please mention how many percentages of water were exchanged each day if you have monitored.

·       For each analysis, please clarify how many fish were taken.

·       Some parts of the discussion are better updated with research in 2021 and 2022 as they refer to some old references. Please update the discussion with the latest studies as much as possible.

·       Although you wrote this section well, you can still improve it by answering these questions and annotated them to the discussion section. Why were these results observed? Discuss more possible reasons.

·       The conclusion needs to be revised and more comprehensive concepts should be added there.

Tables and Figures

•            Please explain a little bit about your experimental diets, per each Table and Figure. Each Table and figure should represent enough information separately from the text.

•            Double-check the units and titles of all Tables.

•            Please mention in the footnote of all Tables which kind of statistical method you used for comparing the means.

Best regards

Author Response

Thank you so much for the reviewer's valuable comments. We have revised our manuscript according to all the comments and suggestions. The following are our comments and responses:

Reviewer #1:

The authors investigated the effect of background color on skin colour variation of juvenile Leopard coral grouper Plectropomus leopardus. They designed five treatments to test their hypothesis. This manuscript (MS) was clearly written and easy to understand. This work can help the sustainability of this species farming as few studies have been done on this topic. However, some minor issues significantly compromised the quality of this MS.

However, I have touched on some more points that can contribute to the improvement of this MS.

Minor comments

  • Line 72, and throughout the MS, please first mention the common name plus scientific name, and for the rest of the MS, just report the common name.

Answer: Thanks for your advice. In this paper, we have done as your suggestion of 'first mention the common name plus scientific name'. And for the rest of the MS, the abbreviated scientific name was used, which conformed to most writing habits.

  • You wrote the next sections very well

Answer: Thank you.

  • Here and elsewhere, report P uppercase and italic (P<0.05).

Answer: Thank you. The P-values in the MS were all uppercase and italic.

  • Throughout the MS, if there is no significant difference, no need to report P-value.

Answer: Thank you. We have revised as your suggestion.

  • Please reorder the keywords alphabetically and capitalize each word.

Answer: Thank you. We have revised as your suggestion.

  • Well-developed introduction and included a clear fellow and relevant points.
  • Please update the introduction with recent works as many studies are available from the last two years, which were not included in this section.

Answer: Thank you. We have revised as your suggestion and updated references.

  • Please mention the novelty of your work in the last paragraph of the introduction.

Answer: Thank you. We have revised as your suggestion.

  • Please mention how many percentages of water were exchanged each day if you have monitored.

Answer: Thank you. We have revised as your suggestion.

  • For each analysis, please clarify how many fish were taken.

Answer: Thank you. We have revised as your suggestion.

  • Some parts of the discussion are better updated with research in 2021 and 2022 as they refer to some old references. Please update the discussion with the latest studies as much as possible.

Answer: Thank you. We have revised as your suggestion and updated references.

  • Although you wrote this section well, you can still improve it by answering these questions and annotated them to the discussion section. Why were these results observed? Discuss more possible reasons.

Answer: Thank you. We have revised as your suggestion.

  • The conclusion needs to be revised and more comprehensive concepts should be added there.

Answer: Thank you. We have revised as your suggestion.

Tables and Figures

  • Please explain a little bit about your experimental diets, per each Table and Figure. Each Table and figure should represent enough information separately from the text.

Answer: Thank you. We have revised as your suggestion.

  • Double-check the units and titles of all Tables.

Answer: Thank you. We have checked the units and titles of all Tables.

  • Please mention in the footnote of all Tables which kind of statistical method you used for comparing the means.

Answer: Thank you. We have revised as your suggestion.

Best regards

Answer: Thank you. Best regards to you.

Reviewer 2 Report

The work is well laid out and written in an adequate level of English.

However, the missing information about water nutrients of the rearing system (comment to Line 120) can potentially overturn the work results. Fish are sensitive to nitrogen compounds and consequently vary their skin color in response to stress.

Line 27 - Please add patronymic to the species in this line and in line 92.

Line 120 – The authors must better describe the rearing system (illumination, filtering, aeration) if the food excess and feces were removed and the chemical parameters as NO2, NO3, NH4, because the fish color is easily influenced by excess of these compounds.

Line 129 - What is the meaning of L*, a*, b*? Add a description.

Line 130 – Why did the authors choose only 3 fish to photograph if 10 were selected?

Line 291- This paragraph is a repetition of the introduction. I suggest deleting it.

Author Response

Thank you so much for the reviewer's valuable comments. We have revised our manuscript according to all the comments and suggestions. The following are our comments and responses:

Reviewer #2:

The work is well laid out and written in an adequate level of English.

Answer: Thank you.

However, the missing information about water nutrients of the rearing system (comment to Line 120) can potentially overturn the work results. Fish are sensitive to nitrogen compounds and consequently vary their skin color in response to stress.

Answer: Thank you. We have revised this paragraph as your suggestion.

Line 27 - Please add patronymic to the species in this line and in line 92.

Answer: Thank you. We firstly mention the common name plus scientific name, then the abbreviated scientific name was used for the rest of the paper.

Line 120 – The authors must better describe the rearing system (illumination, filtering, aeration) if the food excess and feces were removed and the chemical parameters as NO2, NO3, NH4, because the fish color is easily influenced by excess of these compounds.

Answer: Thank you. We have revised this paragraph as your suggestion.

Line 129 - What is the meaning of L*, a*, b*? Add a description.

Answer: Thank you. L* means lightness, a* means redness, and b* means yellowness, we have revised as your suggestion.

Line 130 – Why did the authors choose only 3 fish to photograph if 10 were selected?

Answer: Thank you. Usually, three fish were used to measure the L*, a* and b* values. However, there are some dark spot in the experimental fish skin, which could influence the accuracy of data. Thus, 10 fish were selected to measure L*, a* and b* value, and 3 fish were selected to photograph.

Line 291- This paragraph is a repetition of the introduction. I suggest deleting it.

Answer: Thanks for your advice. But we did not find the repetition.

Round 2

Reviewer 2 Report

Please add informations about NO2 and NO3 values. These nitrogenous composts are fundamental to evaluate the water quality and the subsequently changes in skin colour.

Author Response

Thank you so much for the reviewer's valuable comments. We have revised our manuscript according to the comments and suggestions. The following are our comments and responses:

Reviewer #2:

Please add informations about NO2 and NO3 values. These nitrogenous composts are fundamental to evaluate the water quality and the subsequently changes in skin colour.

Answer: Thank you. we have revised as your suggestion.
